# Severity of Menstrual Pain Is Associated with Nutritional Intake and Lifestyle Habits

**DOI:** 10.3390/healthcare11091289

**Published:** 2023-04-30

**Authors:** Yuna Naraoka, Momo Hosokawa, Satomi Minato-Inokawa, Yuichi Sato

**Affiliations:** 1Intractable Disease Research Center, Graduate School of Medicine, Juntendo University, 2-1-1 Hongo, Bunkyo-ku, Tokyo 113-8421, Japan; 2Japanese Center for Research on Woman in Sport, Juntendo University, 2-1-1 Hongo, Bunkyo-ku, Tokyo 113-8421, Japan; 3General Incorporated Association Luvtelli, 3-2-14, Nihonbashi, Chou-ku, Tokyo 103-0027, Japan; 4Laboratory of Community Health and Nutrition, Department of Bioscience, Graduate School of Agriculture, Ehime University, Matsuyama 790-8566, Japan; 5Department of Pharmacoepidemiology, Graduate School of Medicine and Public Health, Kyoto University, Yoshida-Konoe-cho, Sakyo-ku, Kyoto 606-8501, Japan; 6Obstetrics and Gynecology, Tatedebari Sato Hospital, Takasaki 370-0836, Japan

**Keywords:** menstrual pain, nutrients, lifestyle habits, PMS, Japanese women

## Abstract

Recently, the employment rate of women in Japan has steadily increased. Approximately 80% of women experience menstrual pain and premenstrual syndrome (PMS). These symptoms decrease a woman’s quality of life and her work productivity, leading to an economic loss. This cross-sectional study of 321 healthy Japanese women aged 20–39 years aimed to clarify the lifestyle-related factors or nutrient intake that might cause menstrual pain. The participants underwent body composition measurements and completed meal survey sheets and lifestyle questionnaires, including menstrual status, exercise, sleep and breakfast consumption. Based on the questionnaire results, participants were divided into two groups according to the severity of menstrual pain, namely, heavy and light. Chi-square and Wilcoxon signed-rank sum tests were used to compare the severity of menstrual pain in the two groups. In the heavy group, the intake of animal proteins, including fish, vitamin D and vitamin B12, was significantly lower (*p* < 0.05), as was the frequency of breakfast consumption and bathing (*p* < 0.05). The rate of PMS symptoms was significantly higher in the heavy group (*p* < 0.05). This study suggests that a lack of animal protein, the accompanying vitamins and fatty acids, and the frequency of breakfast or bathing are associated with the severity of menstrual pain.

## 1. Introduction

The employment rate of women in Japan has increased annually to 44.1% [1]. Among the health problems specific to women, dysmenorrhea and premenstrual syndrome (PMS) often reduce the quality of life (QOL) of women of a reproductive age [2]. Symptoms that cause working women to take time off from work related to menstruation include heavy bleeding, abdominal pain, lower back pain and an inability to get out of bed [3,4,5]. According to a systematic review by the World Health Organization in 2006, the prevalence of menstrual pain in women of reproductive age was 16.8–81.0% and with severe symptoms in 12–14% [6]. This prevalence might be higher because of the presence of potential patients without access to clinics and the need for clarity in the pain criteria. On the other hand, the prevalence of menstrual pain in Japanese women is 78.5% [7]. Menstrual pain causes school and work absenteeism and decreases the quality of life [8,9]. In addition, working in poor physical conditions leads to economic losses due to reduced productivity. The estimated annual economic burden extrapolated to the Japanese female population was JPY 683 billion or USD ~8.6 billion [10].Therefore, health problems peculiar to women are not limited to women’s problems, but are issues that need to be addressed immediately by society as a whole.

Dysmenorrhea is defined as pain that starts just before menstruation and lasts during this period. The pain is most severe on the first and second days [11,12]. Menstrual pain is caused by an excessive secretion of prostaglandins, vasopressin and leukotrienes due to uterine contractions [13,14]. The factors that cause these hypersecretions are age, body mass index (BMI), early menarche, prolonged menses, premenstrual syndrome, pelvic inflammation, mental disorder and previous sexual attack [15]. However, this mechanism is intricately intertwined, making it challenging to identify its cause. The general treatment relates to the use of non-steroidal anti-inflammatory drugs (NSAIDs) or oral contraceptives (OCs) [16,17], but taking the drug is a symptomatic treatment. Therefore, it cannot improve the physical condition to prevent pain.

Premenstrual syndrome (PMS) affects around 85% of women who experience some form of symptoms, including mild ones, with 20–25% being moderate to severe and 5% being diagnosed with Premenstrual Dysphoric Disorder (PMDD) [18]. PMS is defined as psychological or physical symptoms occurring mainly during the luteal phase [19], with irritability, tension and depression being the main symptoms, although over 200 types of symptoms have been associated with it [20]. As with menstrual pain, it is a major factor in the decreased quality of life of many women. The main cause is the decrease in serotonin associated with hormones [21]. Treatment involves hormone therapy or medication, and it has been reported that PMS symptoms improve with the intake of oral contraceptives [22]. However, these are also palliative therapies and a fundamental improvement of the constitution is required for improving women’s QOL.

On the other hand, as an alternative therapy, the supplemental intake of nutrients has been shown to relieve menstrual pain or premenstrual syndrome [23]. Previous studies have shown that an intake of 50,000 IU of vitamin D supplements per week for 8–9 weeks [24,25], thiamin at a dosage of 100 mg for 60 days [26], 25,000 IU of vitamin E for 5 days starting 2 days before menstruation [27], and omega-3 polyunsaturated fatty acid at a daily dosage of 2 g fish oil supplement [28,29] improved menstrual pain. In addition, a study on Japanese women reported that dietary fiber reduced menstrual pain [30]. Fathizadeh et al. reported that the physical symptoms were significantly improved compared to placebo when women aged 15–45 diagnosed with PMS took 250 mg of magnesium or magnesium with 40 mg of vitamin B6 for 2 months [31]. Siahbazi et al. reported that PMS symptoms were significantly reduced compared to placebo when women aged 20–35 diagnosed with PMS were given 50 mg of zinc from day 16 of the menstrual cycle to day 2 of the next cycle for 3 months [32]. However, because of lack of evidence [23,33], implementing nutrient intake as an alternative medicine has not yet been established [34,35].

Previous studies have reported on the relationship between lifestyle factors and menstrual pain and menstrual-related symptoms. A survey of university students found that the consumption of cola, meat and alcohol [36] as well as fruits and beans [37] was reported to have an impact on menstrual pain. In addition, a study of Japanese junior high-school students reported that sleep, eating habits, participation in sports clubs and screen time had an impact on the presence of PMS [36,38]. However, most of these studies focused on junior high-school and university students, and there have been few studies on working Japanese women in their 20s and 30s [39]. Furthermore, the relationship between menstrual-related symptoms and diet that has been reported so far has been at the food level, and there are no reports on the relationship between daily nutrient intake and menstrual-related symptoms. However, in order to improve the quality of life of working women, it is important to fundamentally resolve menstrual pain. Therefore, it is important to extract the specific nutrient or lifestyle factors that could cause menstrual pain from lifestyle habits. The objective of this study was to determine the lifestyle-related factors or nutrient intake associated with the severity of menstrual pain in healthy women without gynecological diseases.

## 2. Materials and Methods

### 2.1. Design and Settings

This study was conducted in a cross-sectional study. From 23 May to 6 June 2018, we recruited participants for the study through the online platform of the General Incorporated Association Luvtelli. We investigated selection and exclusion criteria online. The participants who were deemed suitable for the study underwent body composition measurement and completed a dietary survey questionnaire on 8–10 June 2018 at the Wacoal Study Hall (Kyoto, Japan), and the resulting data were used for analysis. The study design was approved by the Sato Hospital Ethical Review Board (protocol code: S201805-01, date: 21 May 2018). All procedures were conducted in accordance with the principles of the amended Declaration of Helsinki. All participants were informed about the possible risks and discomforts involved in the experiment prior to giving their written informed consent to participate.

### 2.2. The Study Sample

We received responses from 511 participants through online recruitment. Among them, 63 were excluded for being 40 years of age or older and for not meeting the inclusion criteria. Additionally, 77 participants who were pregnant or nursing were excluded, leaving 371 potential candidates for the study. On the day of the actual measurement, 325 participants came to the venue, and 4 were excluded due to missing data, resulting in a final study population of 321 participants.

### 2.3. The Participants

The specific inclusion criteria were as follows: healthy women aged 20–39 years. Participants who are using medication (such as non-steroidal anti-inflammatory drugs), had poor lifestyle habits (including eating disorders or alcohol dependence), were in pregnancy or lactating, had a disease and were under treatment (including hormone therapy), were taking oral contraceptives, were participating in other clinical trials during this study, or were deemed inappropriate by a physician’s judgment were excluded from the study. After presenting the explanation document for the study on the online recruitment page, we conducted a survey on selection and exclusion criteria. The 321 participants who were deemed suitable for the study came to the venue and underwent body composition measurements after providing their consent. The dietary survey was answered using a dedicated paper and a pencil. An online questionnaire through Google Forms was used for the survey on menstrual status and lifestyle, which was completed by the participants at the venue.

### 2.4. Measurement of Body Composition

Body weight, fat mass and muscle mass were measured using a Dual Frequency body composition analyzer (DC-430A; TANITA, Tokyo, Japan) and the body mass index was calculated. The participants came to the venue between 10:00 AM and 4:00 PM and after 20 min, underwent measurement in a room with a normal room temperature. The measurement was conducted with clothing on, so the clothing weight was set to 500 g.

### 2.5. The Meal Survey and Lifestyle Questionnaire

The meal survey was conducted using a brief self-administered diet history questionnaire (BDHQ) (Gender Medical Research, Tokyo, Japan) from which the daily intake of carbohydrates, proteins, fats, main vitamins and minerals were calculated. This dietary questionnaire has been validated for comparable nutrient intakes [40,41]. The self-created lifestyle questionnaire consisted of 60 questions in three sections. The first section included sociodemographic data, such as family structure and place of residence. The second section was about menstruation, including menstrual cycle, menstrual pain and PMS. The third section was about lifestyle, including work, sleep, smoking, bathing, exercise and diet. The respondents used their smartphones to answer the questionnaire, which took about 10 min. The questions were created in Japanese and all respondents were native Japanese speakers.

### 2.6. Measures

According to the questionnaire, menstrual pain was investigated using the following four items: “incapacitating pain,” “unable to get through the day without medication,” “pain but does not interfere with daily life” and “almost no pain.” Those who answered “incapacitating pain” or “unable to get through the day without medication” were classified as “heavy,” while those who answered “pain but does not interfere with daily life” or “almost no pain” were classified as “light.”

Regarding PMS, the following symptoms were investigated based on previous studies [42] and were reported to occur from 10 days before menstruation: irritability, becoming easily angered, feeling lethargic, becoming depressed, feeling anxious, decreased concentration, feeling sleepy, feeling easily tired, breast tenderness, abdominal pain, headache, skin problems, coldness in the extremities, swelling, becoming averse to work, becoming averse to being a woman and becoming averse to socializing. Additionally, the presence or absence of symptoms was investigated.

### 2.7. Data Analysis

To eliminate the effects of hormone status or water content, we analyzed 321 non-pregnant and non-lactating women aged 20–39 years with no comorbidities. Nutritional uptake was corrected to the intake of each 1000 kcal because of the difference in meal size. All analyses were performed using SPSS (version 26.0; SPSS, IBM, Armonk, NY, USA), with statistical significance set at *p* < 0.05. The chi-square test was used to compare the ratios between the two groups, and the Wilcoxon signed-rank sum test was used to compare the mean values.

## 3. Results

### 3.1. Background

The mean age, height, weight, BMI, muscle mass and body fat were 30.53 ± 4.69 years, 158.30 ± 6.31 cm, 51.80 ± 7.50 kg, 20.67 ± 2.62 kg/m^2^, 35.18 ± 3.58 kg and 27.02 ± 5.29%, respectively. A total of 84.1% of the participants had a normal menstrual cycle of 25–38 days and 94.4% had normal menstrual periods of 3–7 days (Table 1). The subjects of the study were 321 women, of which 55.7% lived in Kyoto Prefecture, 18.4% in Osaka Prefecture, 5.0% in Hyogo Prefecture, 3.7% in Shiga Prefecture and 17.2% in other regions of Japan. Of these women, 94.1% were employed and 5.9% were unemployed, including housewives. Their occupations consisted of 67.0% desk work, 12.5% standing work with night shifts and 14.6% standing work without night shifts. Married women accounted for 30.8% of the total, while 69.2% were unmarried. In total, 13.4% of the women in the study had children. However, pregnant and lactating women were not included in this research. A total of 321 subjects were analyzed; of these, 76.19% experienced menstrual pain, 1.90% were prostrated with pain, 30.16% could not spend time without medicine, 44.13% had pain that did not interfere with daily life and 23.81% had almost no pain.

### 3.2. Relationship between Menstrual Pain and the Intake of Nutrients

The heavy group had a statistically lower intake per 1000 kcal of total protein (*p* = 0.032), animal protein (*p* = 0.043), vitamin D (*p* = 0.011), vitamin B12 (*p* = 0.017), fish consumed with bones (*p* = 0.019), grilled fish (*p* = 0.008) and dried fish (*p* = 0.020) than the light group. On the other hand, the heavy group had a statistically higher intake of sugar (*p* = 0.041), ramen (*p* = 0.036) and ice cream (*p* = 0.037) (Table 2).

### 3.3. Relationship between Menstrual Pain and PMS

We investigated the PMS symptoms from multiple answers and compared whether there was a difference in the PMS symptoms between the heavy and light groups. Compared to the light group, the heavy group had significantly higher rates of anxiety, somnolence, tiredness, rough skin, cold hands and feet, antisocial behavior (*p* < 0.05), irritability, anger, lethargy, depression, decreased concentration, lower abdominal pain, headache, swelling and a dislike of work (*p* < 0.01) (Table 3). In particular, 48.64% of the light group experienced disconcertedness, while it was 75.25% in the heavy group. In addition, pain-related PMS, such as lower abdominal pain (heavy, 59.41%; light, 25.91%) and headache (heavy, 33.66%; light, 14.55%), was especially severe in the heavy group.

### 3.4. Menstrual Pain and Lifestyle

Regarding lifestyle habits, in the heavy group, 64.4% ate breakfast almost every morning, which is significantly lower than the light group, 73.6% (Table 4). In addition, 26.7% of the heavy group took daily baths, significantly lower than the 40.5% of the light group (chi-square test, *p* < 0.01). There was no significant difference in sleep time between the heavy and light groups, but sleep satisfaction was significantly higher in the light group. Overall, only about 47.4% of the participants exercised more than 30 min a day, and no significant difference was observed, depending on the severity of menstruation.

## 4. Discussion

In this study, we divided the participants into two groups according to the severity of menstrual pain and analyzed their related nutrient intake, lifestyle and PMS symptoms. The heavy group had significantly lower intakes of protein, fish and vitamin B12, D and zinc accompanying fish. In addition, the heavy group had a high frequency of PMS symptoms. In terms of lifestyle, frequency of breakfast intake and bathing habits, the proportion of participants who did not soak in the bathtub and only took a shower was significantly higher in the heavy group. However, a correlation between BMI and menstrual pain was not found in this study.

In this study, the proportion of menstrual pain was 76.19%, which is consistent with the reported proportion in previous studies [6]. Previous studies have examined the characteristics of lifestyle and the presence or absence of menstrual pain. In particular, it has been reported that the group with menstrual pain had a high intake of tea, cola, sugar and meat [38]. In this study, too, it was found that the group with menstrual pain had a high intake of sugar, which is consistent with previous research. However, the results regarding meat intake were different from those of this study. This may be because the original intake of meat was higher in Spain than in Japan and animal fat may be a risk factor, but since there is no analysis of nutrients, the cause is unknown. In previous studies, there was no association between fish intake and menstrual pain, which may be due to differences in dietary habits among Japanese people. In studies targeting Japanese people, it has been reported that the group with mild menstrual pain had a high intake of dietary fiber [30], but no association was found between dietary fiber and menstrual pain in this study. In addition, there have been several reports on the relationship between the frequency of breakfast and menstrual pain [37] or PMS [36,43] in studies targeting middle- and high-school students and university students, but this was also observed in this study targeting working women in their 20s and 30s.

Previous studies have shown that the intake of vitamin D or vitamin B12 supplements reduces pain. Fatemeh et al. revealed that taking 50,000 IU of vitamin D per week for 8 consecutive weeks, regardless of the menstrual cycle, increases the blood levels and alleviates symptoms such as physical pain [25]. In addition, an animal experiment showed that vitamin D metabolism suppresses the production of prostaglandin in the uterine endometrium and restricts its biological activity by affecting the prostaglandin receptors in the endometrium [27,44]. By contrast, vitamin B12 suppressed pain by inhibiting the synthesis of the cyclooxygenase (OCX) enzyme involved in the production of prostaglandins from arachidonic acid in an animal study [28,45]. Furthermore, sodium dextran sulfate-induced colitis showed that a methyl-deficient diet (excluding vitamin B12, folic acid and choline) caused a significant upregulation of COX2 in the gut after exposure to sodium dextran sulfate in rats [29,46]. Vitamin D and vitamin B12 may have potential functions in inhibiting the inflammatory cascade and relieving pain. In the present study, the light group had a significantly high intake of vitamins D and B12. This result suggests that pain may be alleviated by the daily intake of at least the recommended amounts of essential vitamins, as reported in previous studies on anti-inflammatory mechanisms. These results indicated that by properly meeting the necessary nutrient requirements, such as through dietary habits or supplements, menstrual pain can be prevented and improved without using medication. In addition, vitamin D deficiency is known to cause difficulty in becoming pregnant [47,48], and the health conditions of women before pregnancy affects their health transgenerationally [49,50]. Improving menstrual-related symptoms could be a shortcut to maintaining women’s lifelong health as a clear indicator.

The light group also had significantly higher intakes of grilled, dried and raw fish, eaten with bones, suggesting that fish are the source of vitamins D and B12. The consumption of n-3 polyunsaturated fatty acids (PFUA) has also been reported to reduce menstrual pain [29,51]. Savaris et al. reported that the intake of dietary n-3 PFUA from food sources is less favorable in endometriosis patients with pelvic pain than in healthy individuals [52]. However, no significant difference was found between the heavy and light groups regarding n-3 PFUA intake in this study. On the other hand, the heavy group had a higher intake of n-6 PFUA, such as ramen or ice cream. The meal included excess n-6 PFUA, leading to the overexpression of phosphatide on the cell membrane [53]. Diets high in n-6 PFUA significantly increased cell membrane phospholipids. During the menstrual cycle, when progesterone levels decrease, n-6 PFUA, especially arachidonic acid, is released and enters the prostaglandin production cascade, delivering prostaglandins and leukotrienes into the uterus. The inflammatory response is mediated by prostaglandins and leukotrienes, which cause pain, nausea, vomiting, swelling and headaches. In particular, prostaglandins F2α and COX are produced from arachidonic acid and cause vasoconstriction and uterine contraction, which causes uterine ischemia and pain [16]. n-3 PFUA inhibits fatty acid synthases such as Δ6-desaturases (conversion of LA to GLA), elongase (elongation of GLA to DGLA) and Δ5-desaturase (conversion of DGLA to ARA), thereby increasing PG1 and reducing PG2 production [54,55]. This imbalance can lead to menstrual cramps. In this study, n-6 PUFA included in meals progressed prostaglandin synthesis, which might have promoted menstrual pain in the heavy group.

In this study, the heavy group was more likely to develop PMS. Similar to the present study, previous studies reported that the prevalence of PMS was significantly higher in women reporting dysmenorrhea [36,56].A lack of serotonin causes PMS and PMDD [57], whereas a diet lacking the serotonin precursor tryptophan induces PMDD [58]. In our study, the heavy group had a significantly lower protein intake, the primary source of tryptophan, and insufficient vitamin B6, magnesium and vitamin C, which are necessary for serotonin synthesis, causing mental PMS, such as anxiety and irritability. In addition, physical PMS related to pain, such as headache and lumbago, may be associated with the production of prostaglandins and leukotrienes in the same manner as menstrual pain. Prostaglandins and leukotrienes act on uterine contraction, but they also work on the smooth muscles of the stomach, intestines and blood vessels, resulting in headaches and gastrointestinal symptoms [59]. Due to this effect, it is thought that the rate of feeling PMS was higher in the heavy group in this study as well.

Family history has been identified as a factor in menstrual pain and PMS [60,61]. Previous studies have reported that women with a family history of menstrual difficulties, such as their mother or sister, are significantly more likely to have menstrual difficulties compared to women without such a family history. In fact, 90.9% of women with a family history of menstrual difficulties experience similar symptoms, and the risk of menstrual difficulties is three times higher in women with a family history compared to those without [62]. The family history of menstrual pain is known to be inherited in 40–50% of cases [63]. On the other hand, about 56% of information related to menstrual management comes from mothers, and access to medical care is influenced by the maternal education level [64,65]. Since this study did not investigate family history, it is possible that there were undiagnosed endometriosis patients among the participants in this study. However, factors learned from parents, such as lifestyle and dietary habits, may have an impact on the family history of menstrual-related symptoms.

Regarding lifestyle habits, the frequency of eating breakfast, bathing and the degree of satisfaction with sleep were found to be related to menstrual pain, but sleep time and exercise were not associated with pain. In previous research, a relationship between skipping breakfast and menstruation has been reported. In a previous study of Japanese women aged 18 to 20, the group that reported having breakfast 0–3 days per week had a significantly higher incidence of menstrual cycle irregularity and heavy menstrual pain compared to the group that had breakfast 4–7 days per week. However, no relationship was found between breakfast skipping and PMS [37,43]. In this study, the subjects were aged 20 to 39, and the results were consistent with previous research, showing that those who eat breakfast less frequently are more likely to experience menstrual pain.

Breakfast is closely related to body temperature. Ogata et al. found that two meals a day without breakfast for 6 days resulted in a significantly lower core body temperature compared to three meals a day and disrupted the circadian rhythm by altering the peripheral clock genes [66]. In this study, skipping breakfast was significantly more common in the heavy group, possibly because skipping breakfast daily caused a drop in body temperature. However, since body temperature was not measured in this study, the relationship between body temperature and menstrual pain is a subject for future investigation. What is expected from the habit of breakfast and bathing is the improvement in the coldness of the body. Body coldness and pain are related. Prostaglandins push out the uterine blood vessels and contract the uterine muscles to expel the uterine lining from the body. However, when the body is cold, the blood vessels constrict, which may lead to ischemia and hypoxia in the uterine muscle. Anaerobic metabolites accumulated by hypoxia stimulate nociceptors [41]. The congestion of the uterus due to coldness may aggravate pain.

To our knowledge, this study is the first report showing a relationship between differences in nutrient intake from regular meals and menstrual pain or PMS in Japanese women. Previous studies have reported improvements in menstrual pain and PMS with high-dose supplements, but there have been no reports from Japan demonstrating a relationship between differences in nutrient intake from regular meals and menstrual pain or PMS. This study found that nutrients such as protein, vitamin B12, vitamin D and zinc, which are known to improve menstrual pain and PMS, are particularly abundant in fish, and the group with less severe menstrual pain actually had a significantly higher intake of fish. Fish is a main dish in Japanese cuisine [67], and these results support the effectiveness of Japanese cuisine in improving menstrual-related symptoms. The relationship between Japanese cuisine and longevity has been studied previously. It has been suggested that the healthiest and longest-living Japanese diet was in 1975 [68,69], characterized by actively consuming soy products, seafood, fruits, mushrooms and green tea [70]. The results of this study indicate that a Japanese-style diet including fish is not only valuable for longevity, but also for improving menstrual-related symptoms. Furthermore, shifting to a typical Japanese-style diet for Japanese people who have adopted Western-style eating habits may lead to the prevention of menstrual pain.

Our study has some limitations. First, this study calculated the nutrient intake from dietary surveys based on a self-reported dietary recall, which is expected to have errors compared to the actual amount consumed. Second, we did not collect blood, so the relationship between the blood’s nutritional status and menstrual pain is unknown. Third, this study is a survey that is specialized for Japanese people and is influenced by culture and dietary habits. Therefore, it is unclear whether the results will be applicable to other countries and races. In addition, since this is a phenomenological study based on questionnaire surveys, it is not possible to clarify the underlying mechanisms, and further investigation is necessary. However, our data are consistent with a previous study showing that the light menstrual pain group had nutrients that effectively alleviated menstrual pain compared to the heavy menstrual pain group. It is suggested that to prevent menstrual pain that has not been diagnosed as a disease, it is important to consume enough fish-based protein, to moderate the consumption of carbohydrates that contain a lot of n-6 PFUA and to keep the body warm. Preconception care should include these nutrients and lifestyle habits to prevent menstrual pain. Improving menstruation-related symptoms as an easy-to-understand index may be a shortcut to maintaining women’s health throughout their lives. Future research is needed to investigate the relationship between blood prostaglandin concentration, inflammatory markers, pain and related nutrients.

## 5. Conclusions

Our findings suggest that consuming sufficient protein, with a focus on fish, vitamin B12, vitamin D and zinc, as a part of daily meals and adopting lifestyle habits that raise the body temperature, such as having breakfast and taking a warm bath, may be effective in preventing and improving menstrual pain and PMS.

## Figures and Tables

**Table 1 healthcare-11-01289-t001:** Characteristics of the study and control groups.

		Total			Heavy			Light			*p*-Value
		*n* = 321			*n* = 101			*n* = 220			
		Mean ± SD or *n* (%)			Mean ± SD or *n* (%)			Mean ± SD or *n* (%)			
Age	(year)	30.53	±	4.69	30.06	±	4.84	30.78	±	4.61	0.200
Height	(cm)	158.30	±	6.31	158.22	±	5.88	158.13	±	6.54	0.901
Weight	(kg)	51.80	±	7.50	52.01	±	7.63	51.82	±	7.50	0.869
BMI	(kg/m^2^)	20.67	±	2.62	20.77	±	2.82	20.62	±	2.54	0.944
Muscle mass	(kg)	35.18	±	3.58	35.02	±	3.49	35.26	±	3.65	0.337
Body fat	(%)	27.02	±	5.29	27.48	±	5.45	26.80	±	5.24	0.637
Duration of menstrual periods											-
1 to 2 days		1 (0.3)			1 (1.0)			0 (0.0)			
3 to 7 days		303 (94.4)			94 (93.1)			209 (95.0)			
8 to 10 days		14 (4.4)			6 (5.9)			8 (3.6)			
Others		3 (0.9)			0 (0)			3 (1.4)			
Do not know		0 (0.0)			0 (0.0)			0 (0.0)			
Interval of menstrual cycle											-
Under 25 days		12 (3.7)			4 (4.0)			8 (3.6)			
25 to 38 days		270 (84.1)			87 (86.1)			183 (83.2)			
Over 39 days		4 (1.2)			1 (1.0)			3 (1.4)			
Irregular		28 (8.7)			7 (6.9)			21 (9.5)			
Amenorrhea		6 (1.9)			1 (1.0)			5 (2.3)			
Do not know		1 (0.3)			1 (1.0)			0 (0.0)			
Place of residence											-
Urban		209 (65.1)			66 (65.3)			143 (65.0)			
Rural		112 (34.9)			35 (34.7)			77 (35.0)			
Married		99 (30.8)			35 (13.9)			64 (29.1)			-
Have kids		43 (13.4)			14 (13.9)			29 (13.2)			-
Employed		302 (94.1)			92 (91.1)			210 (95.5)			-
Occupation											-
Desk work		215 (67.0)			69 (68.3)			146 (66.4)			
Standing jobs with late-night work		40 (12.5)			9 (8.9)			31 (14.1)			
Standing jobs without late-night work		47 (14.6)			14 (13.9)			33 (15.0)			
Homemaker		19 (5.9)			9 (8.9)			10 (4.5)			

Values are mean ± standard deviation, *p*-value determined using the Wilcoxon signed-rank sum test.

**Table 2 healthcare-11-01289-t002:** Nutritional intake of the study (heavy) and control (light) groups.

		Heavy			Light			*p*-Value	
		*n* = 101			*n* = 220				
		Mean ± SE or *n* (%)			Mean ± SE or *n* (%)				
Energy intake	(kcal/day)	1498.13	±	36.35	1601.75	±	28.43	0.779	
Total protein	(g/1000 kcal)	15.17	±	0.32	15.78	±	0.19	0.032	*
Animal protein	(g/1000 kcal)	8.73	±	0.31	9.36	±	0.19	0.043	*
Plant protein	(g/1000 kcal)	6.43	±	0.11	6.42	±	0.06	0.77	
Fat	(g/1000 kcal)	29.82	±	0.60	29.86	±	0.36	0.718	
Animal fat	(g/1000 kcal)	13.31	±	0.43	13.80	±	0.28	0.436	
Plant fat	(g/1000 kcal)	16.51	±	0.38	16.06	±	0.23	0.495	
Carbohydrate	(g/1000 kcal)	51.05	±	0.81	50.07	±	0.52	0.341	
Iron	(mg/1000 kcal)	4.31	±	0.12	4.50	±	0.07	0.258	
Zinc	(mg/1000 kcal)	4.41	±	0.07	4.56	±	0.05	0.054	
Sodium	(mg/1000 kcal)	2167.98	±	49.66	2205.89	±	26.84	0.092	
Potassium	(mg/1000 kcal)	1333.79	±	36.51	1390.11	±	22.48	0.327	
Calcium	(mg/1000 kcal)	281.15	±	10.38	293.16	±	5.85	0.386	
Magnesium	(mg/1000 kcal)	130.30	±	3.10	136.46	±	1.89	0.135	
Phosphorus	(mg/1000 kcal)	559.26	±	12.45	585.80	±	7.07	0.041	
Copper	(mg/1000 kcal)	0.60	±	0.01	0.61	±	0.01	0.46	
Manganese	(mg/1000 kcal)	1.62	±	0.05	1.52	±	0.03	0.345	
β-carotene	(μg/1000 kcal)	2088.72	±	132.67	2294.40	±	85.80	0.471	
Retinol	(μg/1000 kcal)	394.81	±	21.47	426.91	±	13.36	0.419	
Vitamin D	(μg/1000 kcal)	6.20	±	0.38	7.11	±	0.24	0.011	*
α-tocopherol	(mg/1000 kcal)	4.30	±	0.10	4.42	±	0.06	0.326	
Vitamin K	(μg/1000 kcal)	176.74	±	9.95	186.75	±	5.82	0.358	
Vitamin B1	(mg/1000 kcal)	0.42	±	0.01	0.44	±	0.01	0.14	
Vitamin B2	(mg/1000 kcal)	0.72	±	0.02	0.73	±	0.01	0.82	
Niacin	(mg/1000 kcal)	9.10	±	0.24	9.65	±	0.16	0.048	*
Vitamin B6	(mg/1000 kcal)	0.67	±	0.02	0.71	±	0.01	0.051	
Vitamin B12	(mg/1000 kcal)	4.26	±	0.21	4.84	±	0.14	0.017	*
Folic acid	(mg/1000 kcal)	182.62	±	6.72	187.80	±	3.95	0.83	
Pantothenic acid	(mg/1000 kcal)	3.60	±	0.08	3.69	±	0.05	0.5	
Vitamin C	(mg/1000 kcal)	58.03	±	2.43	58.05	±	1.42	0.407	
Saturated fatty acid	(g/1000 kcal)	9.12	±	0.23	8.99	±	0.14	0.623	
Cholesterol	(mg/1000 kcal)	240.06	±	9.29	249.14	±	6.02	0.206	
Soluble dietary fiber	(g/1000 kcal)	1.68	±	0.05	1.75	±	0.03	0.522	
Insoluble dietary fiber	(g/1000 kcal)	4.46	±	0.13	4.66	±	0.08	0.419	
Total dietary fiber	(g/1000 kcal)	6.34	±	0.19	6.65	±	0.11	0.359	
Salt	(g/1000 kcal)	5.48	±	0.13	5.57	±	0.07	0.095	
Sugar	(g/1000 kcal)	3.43	±	0.19	2.99	±	0.10	0.041	*
Alcohol	(g/1000 kcal)	2.74	±	0.56	3.07	±	0.34	0.392	
Daidzein	(mg/1000 kcal)	7.54	±	0.57	8.00	±	0.34	0.578	
Genistein	(mg/1000 kcal)	12.82	±	0.96	13.61	±	0.58	0.573	
n-3 fatty acid	(g/1000 kcal)	1.31	±	0.04	1.38	±	0.03	0.108	
n-6 fatty acid	(g/1000 kcal)	5.72	±	0.13	5.75	±	0.08	0.502	
n-6/n-3 ratio	-	0.23	±	0.00	0.24	±	0.00	0.053	
Green tea	(g/1000 kcal)	104.86	±	153.35	69.09	±	104.01	0.184	
Red tea and Oolong tea	(g/1000 kcal)	51.78	±	77.74	45.39	±	80.09	0.157	
Coffee	(g/1000 kcal)	94.48	±	103.69	89.19	±	101.51	0.809	
Fish (consumed with bones)	(g/1000 kcal)	3.49	±	6.29	4.11	±	5.75	0.019	*
Grilled fish	(g/1000 kcal)	14.01	±	13.46	17.42	±	14.28	0.008	**
Dried fish	(g/1000 kcal)	6.38	±	7.50	7.89	±	7.92	0.020	*
Squid, octopus, shrimp, shellfish	(g/1000 kcal)	5.82	±	4.79	7.06	±	5.69	0.078	
Ramen	(g/1000 kcal)	8.06	±	9.10	6.42	±	8.29	0.036	*
Udon	(g/1000 kcal)	10.54	±	8.98	12.47	±	10.18	0.361	
Pasta	(g/1000 kcal)	9.32	±	10.45	9.73	±	9.44	0.322	
Ice cream	(g/1000 kcal)	15.56	±	17.61	12.05	±	14.54	0.037	*
Western confectionery	(g/1000 kcal)	23.55	±	18.30	20.51	±	17.57	0.087	
Japanese confectionery	(g/1000 kcal)	3.53	±	5.03	3.60	±	3.61	0.204	

Values are mean ± standard deviation, *p*-value determined using the Wilcoxon signed-rank sum test. * *p* < 0.05, ** *p* < 0.01.

**Table 3 healthcare-11-01289-t003:** The distribution of PMS in menstrual pain severity.

	Heavy (*n* = 101)		Light (*n* = 220)		*p*-Value	
	*N*	%	*N*	%		
Disconcertedness	76	75.25	107	48.64	<0.001	**
Anxiety/irritability	51	50.50	69	31.36	0.001	**
Anemia	29	28.71	33	15.00	0.004	**
Despressed mood	54	53.47	64	29.09	<0.001	**
Anxiety tension	32	31.68	47	21.36	0.046	*
Difficalty concentrating	36	35.64	36	16.36	<0.001	**
Hypersomia	73	72.28	128	58.18	0.015	*
Fatigue/lack of energy	51	50.50	80	36.36	0.017	*
Chest pain	42	41.58	67	30.45	0.051	
Abdominal pains	60	59.41	57	25.91	<0.001	**
Headache	34	33.66	32	14.55	<0.001	**
Skin problem	51	50.50	81	36.82	0.021	*
Cold limb	11	10.89	10	4.55	0.033	*
Swelling	35	34.65	44	20.00	0.005	**
Depressed interest in work	23	22.77	25	11.36	0.008	**
Depressed by myself as a woman	20	19.80	8	3.64	0.055	
Depressed interest in social activities	15	14.85	15	6.82	0.022	*
Without symptoms	15	14.85	14	6.36	0.035	*

Values are mean ± standard deviation, *p*-value determined using the chi-square test. * *p* < 0.05, ** *p* < 0.01.

**Table 4 healthcare-11-01289-t004:** Lifestyle habits and menstrual severity.

	Total (*n* = 321)		Heavy (*n* = 101)		Light (*n* = 220)		*p*-Value	
	*N*	%	*N*	%	*N*	%		
Frequency of eating breakfast							<0.001	**
Almost everyday	227	(70.7)	65	(64.4)	162	(73.6)		
4 or 5 times a week	36	(11.2)	13	(12.9)	23	(10.5)		
2 or 3 times a week	25	(7.8)	7	(6.9)	18	(8.2)		
Once a week	14	(4.4)	7	(6.9)	7	(3.2)		
No breakfast	19	(5.9)	9	(8.9)	10	(4.5)		
Frequency of taking a bath							<0.001	**
Almost everyday	116	(36.1)	27	(26.7)	89	(40.5)		
4 or 5 times a week	46	(14.3)	14	(13.9)	32	(14.5)		
2 or 3 times a week	43	(13.4)	18	(17.8)	25	(11.4)		
Once a week	36	(11.2)	12	(11.9)	24	(10.9)		
Only taking a shower	80	(24.9)	30	(29.7)	50	(22.7)		
Sleep duration							0.091	
Under 4 h	2	(0.62)	2	(2)	0	(0)		
4–5 h	27	(8.41)	12	(12)	15	(6.8)		
5–6 h	109	(34)	29	(29)	80	(36)		
6–7 h	134	(41.7)	45	(45)	89	(40)		
7–8 h	42	(13.1)	10	(9.9)	32	(15)		
Over 8 h	7	(2.18)	3	(3)	4	(1.8)		
Sleeping certification							0.011	*
Contentedness	133	(41.4)	53	(52)	80	(36)		
Normal	171	(53.3)	46	(46)	125	(57)		
Dissatisfactory	17	(5.3)	2	(2)	15	(6.8)		
Frequency of activities more than 30 min a day							0.629	
4 times a week	8	(2.49)	4	(4)	4	(1.8)		
2–3 times a week	40	(12.5)	13	(13)	27	(12)		
Once a week	47	(14.6)	12	(12)	35	(16)		
1–3 times a month	57	(17.8)	16	(16)	41	(19)		
Not at all	169	(52.6)	56	(55)	113	(51)		

Values are mean (%), *p*-value determined using the chi-square test. * *p* <0.05, ** *p* <0.01.

## Data Availability

The data analyzed during this study are included in this published article. Further inquiries can be directed to the corresponding authors.

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
