# Peer review of "Severity of Menstrual Pain Is Associated with Nutritional Intake and Lifestyle Habits"

_healthcare, 2023, doi:10.3390/healthcare11091289_

Round 1

Reviewer 1 Report

In my opinion the paper can be informative and provide a valuable source document for anyone requiring a primer to know and understand this issue. But, numerous shortcomings in the sections Introduction, Objectives, Methods, Discussion and Conclusions make this paper not appropriate for publication in this form and significant corrections should be made (major revision). Some comments:             

  • Line 35: Cite the appropriate reference for this sentence.  

  • Lines 35-37: Cite the appropriate reference in this sentence.     

  • Lines 68-69: Align the aim of the study stated in this sentence with the aims of the study stated in Abstract (Lines 20-22: `This study cross-sectional study of 321 healthy women aged 20–39 years aimed to clarify the nutrient intake that might cause menstrual pain and PMS.`).   

  • Line 81: If the objectives of the study include premenstrual syndrome, add a new paragraph in which the authors will provide a definition for premenstrual syndrome, as well as relevant data from the literature as mentioned above for menstrual pain.      

  • Line 82: Add subsection `Study Settings`.      

  • Lines 84-86: It is stated that the recruitment and selection of participants in this study was conducted online. Specify the way in which the survey was conducted and measurement of body composition of the participants in this study: Whether the survey (meal survey, etc) was online or via paper and pencil.       

  • Line 97: Add paragraph `Study Sample', with a detailed description (number of participants in the study, participation rate, response rate, etc).  

  • Lines 98-102: Specify the study design. It is not enough to say `observational study', because there are numerous types of observational studies.      

  • Line 102: Add subsection `Measures` in which the way the variables `menstrual pain` and `premenstrual syndrome` were determined will be described. Specify which measures or questionnaires were used for determining those variables. State how the classification of severity of menstrual pain was carried out.        

  • Lines 103-105: Specify where and under what conditions the `Measurement of body composition' was performed.     

  • Lines 106-112: Specify the location where the meal survey was conducted. Provide a reference for the diet history questionnaire. State whether the diet history questionnaire has been validated, with an appropriate reference.     

  • Lines 114-115: Specify `lactating women`. Check the criteria for inclusion and exclusion in this study listed in the subsection ``Participants'' above and match them.    

  • Lines 118-119: Give an explanation of what `the two groups' are in the Method section. State the criteria for classifying participants into those two groups.   

  • Lines 121-133: Section Introduction in this manuscript has described in great detail the importance of menstrual pain and premenstrual syndrome in working women. To that end, present data on the socio-demographic characteristics of the participants in this study: place of residence, occupation, education level, employment, marital status, pregnancy, children).    

  • Line 140: Insert `Table 2`.     

  • Lines 144-145: `Outcome/Outcomes' in this study must be specified.      

  • Line 159: Correct to `Table 3`. And check the others.       

  • Line 186: Section Discussion as a whole should be corrected. The main drawback of the Discussion section is that no comparison of the results of this study with the results obtained in other similar studies is presented. Section Discussion was reduced only to the explanation of the results obtained in this study. Correct in such a way as to remove the stated deficiency.   

  • Lines 280-283: Improve the conclusion in light of the results presented.      

Reviewer 2 Report

Nicely written paper. A good research idea 

However, some areas may need more information 

Please elaborate briefly on the General Incorporated Association Luvtelli, Japan website.  Is this a company? If it is, what kind of business does it do and why has the researcher selected this company

Exclusion criteria: Should include women on hormonal therapy for contraception purposes or other purposes other than pain

Please provide the ID for the Sato Hospital Ethical Review Board and Research Ethics Committee, Faculty of Medicine and the date of approval

Please elaborate on the questionnaires used for lifestyle and menstrual status- Number of sections, the title of each section, the language used in the questionnaire, the number of questions, how long it took one participant to complete it etc….

Well-written discussion! 

Reviewer 3 Report

Please find the attached document.

Reviewer 4 Report

The paper investigates the potential relationship between nutrient intake, lifestyle factors, and menstrual pain and premenstrual syndrome (PMS) in Japanese women aged 20-39 years. It employs a cross-sectional study design, which examines a group of individuals at a specific point in time. While the study's sample size of 321 healthy women is relatively small, its findings suggest that there may be a link between a lack of animal protein, accompanying vitamins and fatty acids, and the frequency of breakfast or bathing with the severity of menstrual pain. The heavy group exhibited significantly lower intake of animal proteins, including fish, vitamin D, and vitamin B12, as well as less frequent breakfast consumption and bathing, all of which were associated with a higher rate of PMS symptoms.

 The paper's results have the potential to inform public health initiatives and nutritional education by providing healthcare providers with useful information for advising women about dietary and lifestyle changes that may help alleviate menstrual pain and PMS symptoms. However, the study has some limitations, including the cross-sectional design and the use of self-reported measures, which may introduce recall bias.

 Overall, the paper provides a valuable contribution to the topic on menstrual pain and PMS, highlighting potential associations with nutrient intake and lifestyle factors. Nevertheless, further research is necessary to confirm and expand on these findings, as well as to investigate underlying mechanisms.

Specific comments:

1. There is a need of justification of the novelty of the research. There are also other studies in this field which must be better illustrated in part of introduction.

2. The introduction style must be changed. It would be beneficial to highlight and add investigations concerning the relationship between diet, menstrual pain, and other menstrual characteristics, which could provide additional context and relevance to the current study's findings. This will provide also the difference and novelty of current investigation. 

3. The section of materials and methods needs to be changed and better described. Especially the selection of the methods. The use of self-reported measures, which may introduce recall bias must be explained because it has a strong link with the final outcome and obviously provide a precision of the investigation.

4. The limitations of the study must be added. 

Reviewer 5 Report

The authors present an interesting study on the impact of nutrient intake on menstrual pain and PMS. 

1. Can the authors discuss the impact of family history of PMS or menstrual pain?

2. Did the authors consider the endometriosis as a cause of increased menstrual pain?

Reviewer 6 Report

The study attempted to answer the question of whether lifestyle factors and/or the amount of nutrient intake may limit the intensity of menstrual pain in healthy Japanese women.

In the opinion of the reviewer, the work does not bring new knowledge in this area, because this type of research has already been conducted, the more so that the results obtained are comparable to the results of previously conducted research, to which the authors refer in the text.  An original element of the work is the inclusion in the study of food products and dishes characteristic of the diet of the Japanese population.

Research shows that consuming enough protein, mainly fish, in  daily diet and raising your body temperature can be effective in preventing and relieving menstrual pain. Results can be used as a basis for developing a model of nutrition for women with severe menstrual pain.

While the analysis of women's diet and the assessment of the supply of individual nutrients is justified, it is incomprehensible for the reviewer to study the frequency of eating only breakfast. Why were other meals not included in the studies?

The conclusions result from the conducted research and constitute the answer to the questions posed, which are the subject of the work. The literature is well selected and correctly cited in the text of the publication.

Round 2

Reviewer 1 Report

Thank you for the opportunity to re-review manuscript ID: healthcare-2256288. Overall, the authors submitted a version of the manuscript for re-review in which significant corrections were made: the revised manuscript is satisfactorily clear and informative for the topic it describes. The authors satisfactorily answered all my comments and provided appropriate explanations.

In more details, by sections:    

  • In the Introduction section, significant additions were made, starting with definitions and corresponding references, to the much more clearly presented goals of this work.  
  • The section Methods substantially supplemented with data about `Study Settings`, `Study design`, `Study sample`, eligibility criteria, surveys, data collection, etc.       
  • The Results section was corrected in the revised version of this manuscript (including the detailed data for characteristics of the study participants and a detailed description of nutritional intake, as well as for association of severity of menstrual pain with nutritional intake and lifestyle habits).    
  • The Discussion section is substantially improved, with a detailed comparison of the results of this study with the results of similar studies in the world, with the provision of appropriate explanations for the described differences between the studies and the interpretation of the obtained findings. The Discussion section has been improved by introducing a significant number of relevant references in the revised version of the manuscript.  
  • The Conclusions reflect the results presented in the manuscript.  

I thank the authors. Good job!   

Reviewer 4 Report

Significant changes are done. The paper can be published in a present form.